# Unveiling Encoder-Free Vision-Language Models

**Haiwen Diao**[1,2*] **Yufeng Cui**[2*] **Xiaotong Li**[3,2] **Yueze Wang**[2] **Huchuan Lu**[1†] **Xinlong Wang**[2†]

[1]Dalian University of Technology [2]Beijing Academy of Artificial Intelligence [3]Peking University

diaohw@mail.dlut.edu.cn, yfcui@baai.ac.cn, lixiaotong@stu.pku.edu.cn

yzwang@baai.ac.cn, lhchuan@dlut.edu.cn, wangxinlong@baai.ac.cn

Code & Models: https://github.com/baaivision/EVE

## Abstract

Existing vision-language models (VLMs) mostly rely on vision encoders to extract visual features followed by large language models (LLMs) for visual-language tasks. However, the vision encoders set a strong inductive bias in abstracting visual representation, *e.g.*, resolution, aspect ratio, and semantic priors, which could impede the flexibility and efficiency of the VLMs. Training pure VLMs that accept the seamless vision and language inputs, *i.e.*, without vision encoders, remains challenging and rarely explored. Empirical observations reveal that direct training without encoders results in slow convergence and large performance gaps. In this work, we bridge the gap between encoder-based and encoder-free models, and present a simple yet effective training recipe towards pure VLMs. Specifically, we unveil the key aspects of training encoder-free VLMs efficiently via thorough experiments: (1) Bridging vision-language representation inside one unified decoder; (2) Enhancing visual recognition capability via extra supervision. With these strategies, we launch **EVE**, an encoder-free vision-language model that can be trained and forwarded efficiently. Notably, solely utilizing 35M publicly accessible data, **EVE** can impressively rival the encoder-based VLMs of similar capacities across multiple vision-language benchmarks. It significantly outperforms the counterpart Fuyu-8B [3] with mysterious training procedures and undisclosed training data. We believe that **EVE** provides a transparent and efficient route for developing pure decoder-only architecture across modalities.

## 1  Introduction

Recently, significant advancements in Large Language Models (LLMs) have catalyzed the emergence of Vision-Language Models (VLMs), showcasing powerful visual perception and cognition capability in visual question answering, image captioning, world knowledge understanding, *etc*. Typically, vision encoders (*e.g.*, CLIP [61] and EVA [22]) focus on extracting highly-compressed visual-semantic representations, succeeded by adaptable language models (*e.g.*, LLaMA [75] and Vicuna [10]) to handle vision-language alignments and instruction-following requirements. Nevertheless, these encoder-based VLMs have several potential drawbacks as shown in Figure 1:

(1) Image Resolution / Aspect Ratio. Existing LVMs are pre-trained with square and fixed-size images. However, this restriction forces VLMs to resize, pad, or partition images of varying shapes [48, 84, 51, 20], resulting in large layout distortion, fragmented connection between image slices, and extra computational burden [81], especially in terms of high-resolution images.

(2) Deployment Overhead. Generally, LVMs and LLMs are executed successively. The growing scale of LVMs [8, 73, 13] severely undermines computational efficiency in real-world deployment, especially when high-resolution images are divided and processed through LVMs multiple times.

---

[*]Equal contribution. [†] Correspondence to *lhchuan@dlut.edu.cn, wangxinlong@baai.ac.cn*.

38th Conference on Neural Information Processing Systems (NeurIPS 2024).

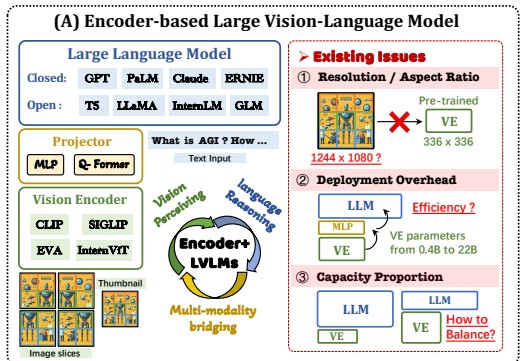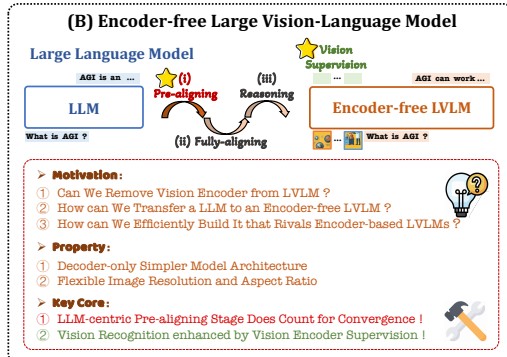

Figure 1: Overview of large (A) encoder-based and (B) encoder-free vision-language models. **Encoder-based VLMs** contain vision encoders (VE) and large language models (LLM), with a projector as a vision-language bridge, while **encoder-free VLMs** exclude vision encoders and handle vision perception and linguistic instruction simultaneously with one unified architecture.

(3) Model Capacity between LVMs and LLMs. Existing LVMs and LLMs are compartmentalized during the pre-training stage, raising uncertainty about how to match their capacities and capabilities. As the scale of LLMs rises from 1.3B to more than 540B, how to strike corresponding vision encoders to maximize their respective abilities remains tricky and elusive.

Given the above considerations, we wonder that: *Is it possible to bypass the constraints of vision encoders and integrate perception and reasoning capabilities into a single unified architecture?* Nevertheless, we empirically discover that even with enormous training data and large model sizes, this conceive remains highly challenging, suffering from greatly slow convergence and large performance gaps compared to encoder-based VLMs (*e.g.*, Fuyu-8B [3] vs. LLaVA-1.5 [49]). The essential problems of constructing encoder-free VLMs from scratch stem from:

**(1) Representation Unity and Emergent Ability.** Due to lacking high-quality image-text data, directly training encoder-free VLMs from scratch is impractical and expensive to learn abundant knowledge and universal representations across modalities. In contrast, the industry has capitalized on the wealth of language data and continuously updated various LLMs [75, 60, 5] with impressive capabilities. Consequently, we position LLMs as a central pivot and strive to compel LLMs per se to develop visual perception while preserving original linguistic proficiency. The process is not a light breeze. We observe that before scaling up pre-trained data, vision-language pre-aligning from an LLM-centric perspective does count. This dramatically prevents model collapse and optimization interference, leading to rapid yet stable model convergence and great performance gains.

**(2) Visual Recognition Capability.** Contrastive learning [61, 62], masked image modeling [45, 86] and auto-regressive generation [21, 16, 19] essentially attempt to prompt visual backbones to produce highly compressed holistic semantics and frequently neglect fine-grained visual clues [14, 15, 17, 18]. On the contrary, we transmit visual signals almost losslessly into encoder-free VLMs. This operation can allow VLMs to autonomously acquire the necessary visual-semantic information. By aligning with patch features captured from vision backbones and text labels predicted by encoder-based VLMs, we implicitly embed visual encoders with input-restricted patterns into encoder-free VLMs for comprehensive visual perception and cognition. This also sidesteps the expensive re-training process of visual encoders for arbitrary image shapes inside encoder-based VLMs.

From the above perspective, we launch **EVE-7B**, an encoder-free VLM evolved from Vicuna-7B [10] and trained with two 8-A100 (40G) nodes in ~9 days. The encoder-free property naturally supports high-resolution images with arbitrary aspect ratios. Notably, solely utilizing 35M publicly accessible data, we efficiently construct a decoder-only VLM that can impressively rival the encoder-based VLMs of similar capacities across multiple vision-language benchmarks. More importantly, we are pioneering a transparent, efficient, and practical route for subsequent VLM research. Our current version significantly outperforms the counterpart Fuyu-8B [3], which relies on mysterious training procedures and undisclosed training data. Beyond facilitating the transition from LLMs to VLMs, EVE holds promising potential for developing scalable and efficient training strategies for encoder-free multi-modality models, requiring fewer training data and device resources.

## 2 Related Work

### 2.1 Encoder-based Vision-Language Model

With the remarkable advancements in large language models (LLMs) [75, 60, 1, 24, 76] and large vision models (LVMs) [61, 22, 73, 8, 13], integrating LLMs with LVMs has become mainstream for building vision-language models (VLMs) effectively. Commercial VLMs extend the capabilities of their proprietary LLM to incorporate images, texts, audio, and videos, including Anthropic's Claude-3V series [1], StepFun's Step-1V [68], xAI's Grok-1.5V [79], Apple's MM1 [58], Google's Gemini series [74], and OpenAI's GPT-4V [83]. In terms of open-source VLMs, existing methods (*e.g.*, BLIP series [42, 43, 12], LLaVA series [50, 49, 51], Emu series [72, 70], Intern-VL [8, 9], and *etc*) have demonstrated promising performance by employing simple intermediate layers to bridge the gap between LVMs and LLMs. Recently, some studies [48, 49, 20, 28] have recognized the significance of input image resolution and aspect ratio for visual perception and cognition, particularly in the interpretation of document, chart, table, and infographic data. However, limited by pre-trained resolution, vision encoders are compelled to partition images into multiple slices or explore a dual-path architecture for low-resolution and high-resolution images respectively, resulting in significant image distortion, fragmented relationship between image slices, and additional computational consumption. For model deployment, some popular open-source libraries, *e.g.*, SGLang [91] or vLLM [39] already support inference acceleration for auto-regressive prediction of encoder-based VLMs. However, as the capacity of vision encoders scales up, the deployment efficiency of vision models would increasingly become a bottleneck for encoder-based VLMs. Meanwhile, how to match the capacities and capabilities between LVMs and LLMs for encoder-based VLMs remains a highly debatable problem with no definitive conclusion. Some studies [49, 51] highlight the notable benefits via substituting CLIP-ViT-B with stronger CLIP-ViT-L-336px in enhancing multimodal models alongside Vicuna-7B [10]. Conversely, other findings [65] indicate that larger vision encoders may not be necessary, as features of multi-scale smaller ones can approximate their performance. Moreover, recent state-of-the-art approaches [9, 70] exhibit significant performance improvements by introducing extremely LVMs. Based on these observations, we attempt to explore a pure decoder-only VLM excluding vision encoders and integrate vision-language understanding and reasoning capabilities into one unified architecture. This could effectively bypass the inherent problems inside encoder-based VLMs, including input constraints of pre-trained vision encoders, inefficiency issues of application deployment, and tricky capacity trade-offs between LVMs and LLMs.

### 2.2 Encoder-free Vision-Language Model

Fuyu-8B [3] stands out as a pure decoder-only network that processes image inputs without relying on an image encoder. Fuyu-8B's design naturally handles high-resolution images with arbitrary aspect ratios, because image patches are fed directly into the model through a simple linear projection layer. However, it demonstrates only average performance across vision-language benchmarks and lacks transparency in training strategies and data sources. This straightforward architecture has inspired further research [40, 31, 44], which focuses on developing powerful supervised instruction datasets to further enhance application capabilities. In response, we explore a practical and promising direction toward developing pure VLMs and breaking the obstacles between encoder-based and encoder-free VLMs. We reveal two crucial lessons: (1) Before scaling up pre-trained data, it is essential to prioritize vision-language pre-alignment from an LLM-centric perspective. This foundational step stabilizes the training process and alleviates optimization interference for integrating visual and linguistic information. (2) Enhancing image recognition capability via visual representation supervision and language conceptual alignment generates stronger visual representations for various vision-language tasks. With publicly available web data, our strategies efficiently accelerate model convergence and achieve performance that can surpass Fuyu-8B and match encoder-based VLMs.

## 3 Methodology

### 3.1 Model Architecture

Figure 2 illustrates the overall framework of our proposed EVE. Firstly, we initialize decoder-only EVE by Vicuna-7B [10] to acquire rich linguistic knowledge and strong instruction-following ability. Under the premise of encoder-free VLMs, we then construct a lightweight patch embedding layer and

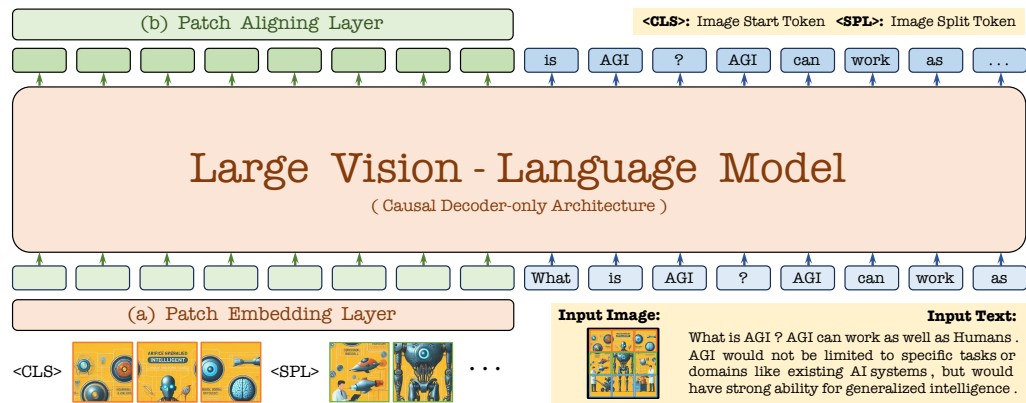

Figure 2: Overview of our proposed EVE. We start by encoding images using a concise (a) patch embedding layer, and then concatenate the patch and word features into a decoder-only network. Next, we facilitate image perception through visual representation supervision using a (b) patch aligning layer, and achieve linguistic conceptual alignment using a next-word prediction pretext task.

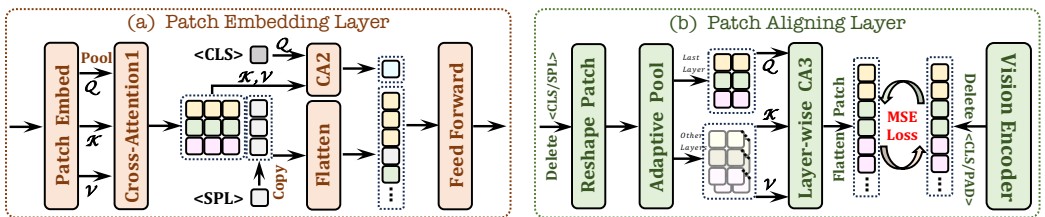

Figure 3: Overview of (a) patch embedding and (b) patch aligning layer. The former layer encodes images into patch features and employs cross-attention (CA1) within a limited receptive field to enhance representations. Meanwhile, a special <CLS> token provides a holistic view of each patch feature in the subsequent backbone. The latter layer removes all meaningless tokens and adjusts the size of patch features to align with the semantics of the same positional features in the vision encoder.

encode image and text inputs efficiently into the above network. After obtaining the network outputs, we attempt to align patch features with pair-wise ones from the vision encoder (VE) [61] through a hierarchical patch aligning layer. Meanwhile, EVE predicts next-word labels that are generated by multi-source encoder-based VLMs via Cross-Entropy (CE) loss.

**Patch Embedding Layer (PEL).** We introduce a succinct and trainable structure in Figure 3(a) to transmit images almost losslessly, rather than using deep encoders or tokenizers to compress image information into high-level semantic representations. Given an image with $(H, W)$ resolution, we first apply a convolution layer to obtain a 2-D feature map with $(h, w)$ sizes. To flexibly control computational overhead, we then implement an average pooling layer within each uncrossed feature slice. To further enhance these downsampled features, a Cross-Attention (CA1) layer is employed in a limited receptive field between each resulting feature and its corresponding slice. Besides, we employ a Cross-Attention (CA2) layer between a special <CLS> token and all patch features. The obtained feature serves as the starting symbol of the image and provides holistic information for patch features in the subsequent backbone. Considering the varying aspect ratios of image inputs, we insert a learnable newline <SPL> token at the end of each row of patch features. This helps the network understand the 2-D spatial structure and dependencies of the image. Finally, we flatten these features and pass them through a two-layer Feed Forward layer, which are then concatenated with text embeddings into one unified decoder-only architecture. Note that the aspect ratios of input images can be arbitrary, requiring no predefined sets, absolute position embedding, or partitioning operations to accommodate a pre-trained vision encoder.

**Patch Aligning Layer (PAL).** Besides coarse-grained text supervision, we further facilitate fine-grained representations by learning from the pre-trained vision encoder. Moreover, it is challenging to establish a shared space for aligning visual features with the vision encoder's output and mapping

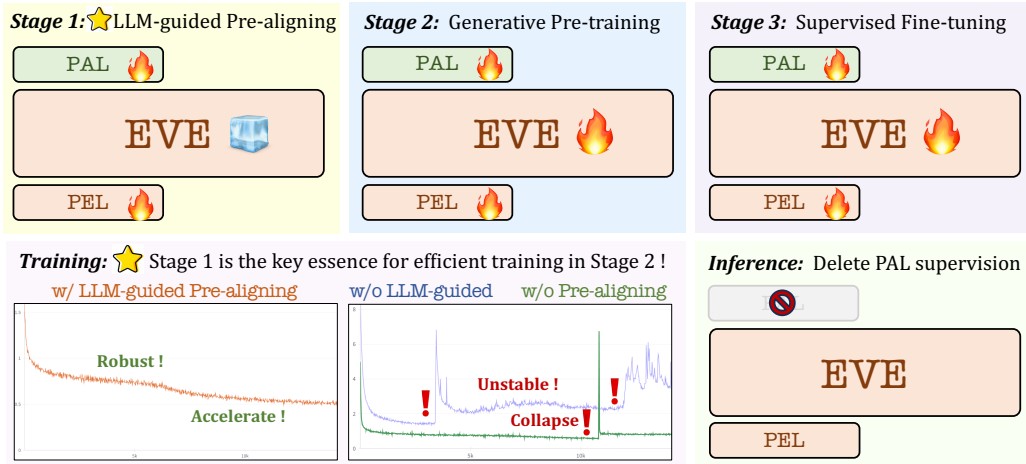

Figure 4: Overview of training procedure with three successive stages. We perform initial vision-language alignment guided by a frozen LLM in Stage 1, and then update the entire backbone for Stage 2 and Stage 3. We empirically find that Stage 1 is quite crucial to avoid collapse and accelerate convergence, thereby enhancing training efficiency. Notably, PAL is removed during inference.

text features to language vocabulary simultaneously. Hence, we explore a hierarchical aggregation strategy in Figure 3(b) to integrate intermediate features across $l$ layers from entire $L$ layers (Interval $= \frac{L}{l}$). Specifically for the vision encoder, we discard meaningless <CLS>/<PAD> tokens from its final output and record the 2-D shape of valid patch fields. For EVE, we first exclude <CLS>/<SPL> tokens from intermediate features of selected layers, and reshape the sequential features back into their original 2-D shape, each of which aligns with the previously recorded shape from the vision encoder via an adaptive pooling layer. We then implement a layer-wise Cross-Attention (CA3) function, using tokens from the last layer as the Query and corresponding positional tokens from other layers as the Key and Value. We normalize the token features obtained from multi-layer aggregation to better match the normalized one from vision encoder one-to-one, utilizing Mean Squared Error (MSE) loss. Such an operation "implicitly" compresses a vision encoder with absolute position embedding (small resolution, fixed aspect ratio) into the decoder-only EVE framework (flexible resolution, arbitrary aspect ratio), enhancing the visual perception ignored by overly simplistic captions.

## 3.2 Training Procedure

In Figure 4, the entire training process is categorized into three successive stages, where we train EVE with publicly available image data captioned by existing VLMs, diverse QA data, and multi-modality dialogue datasets. Notably, after finishing training, we remove PAL supervision during inference.

**LLM-guided Pre-training.** The primary goal of this stage is to establish an initial connection between vision and language modalities, thereby driving LLM to understand the visual concepts and entities in the images from scratch. We introduce publicly available web-scale data in Table 1, including image-only: SA-1B [36], OpenImages [38]; and image-text: LAION [63]. We remove noisy text captions and reproduce 33M high-quality descriptions via Emu2 (17B) and LLaVA-1.5 (13B) as EVE-cap$_{33M}$. Here, only patch embedding and aligning layers are trainable to align with frozen Vicuna-7B [10] for better initializing an encoder-free VLM. Considering that these layers have limited parameter capacities and capabilities, we only adopt 16M of 33M image-text data (EVE-cap$_{16/33M}$) in this stage. We minimize CE loss with text labels and patch-wise MSE loss between EVE and vision encoder. We empirically discover that Stage 1 does count for efficient training, as it prevents collapse and accelerates convergence throughout the entire process.

**Generative Pre-training.** In this phase, we explore in-depth pretraining strategies by unfreezing patch embedding and aligning layers, and the full LLM modules for comprehending vision-language contents. Here, we make use of all 33M image-text pairs (EVE-cap$_{33M}$), and keep both text CE loss and image MSE loss as training objectives. Nevertheless, we discover that though multi-modality performance gradually increases, language capability suffers from a significant downtrend. To strike

Table 1: Details of pre-training datasets. We only extract 16/33M data for Stage 1, and utilize the full 33M data for Stage 2

| Dataset | Captioner | #Samples |
|---|---|---|
| SA-1B [36] | LLaVA-1.5 (13B) | 11M |
| OpenImages [38] | Emu2 (17B) | 7M |
| LAION [63] | Emu2 (17B) | 15M |
| EVE-cap$_{33M}$ | Total | 33M |

Table 2: Details of fine-tuning datasets. We only adopt LLaVA-mix-665K for standard EVE-7B, and further train EVE (HD) with high resolution and all datasets

| Task | Dataset | #Samples |
|---|---|---|
| OCR-VQA | AI2D [33], Synthdog [34], DVQA [32], ChartQA [57], DocVQA [11] | 270K |
| Multi-Task | Vision-Flan [82], Bunny-695K [27] LLaVA-mix-665K [49] | 1.5M |

a compromise between enhancing vision-language capacities and maintaining linguistic competency, we opt for a relatively lower learning rate for stable optimization. Alternatively, involving language-only data remains a prospective avenue for further exploration [53]. Stage 2 serves as a catalyst, driving EVE per se to develop vision-language encoding and aligning capabilities.

**Supervised Fine-tuning.** After obtaining a well-aligned state, EVE is expected to possess further capabilities for following linguistic instructions and learning dialogue patterns. The entire architecture is updated using the same loss functions in Stage 2. Here, we utilize LLaVA-mix-665K [49] (LLaVA-mix$_{665K}$) with both VL and pure NLP dialogues as supervised fine-tuning (SFT) data to obtain the standard version of EVE-7B. Besides, we also attempt to enlarge the limitation of maximum resolution *only* in the SFT stage. To bridge the resolution gap between pre-training and fine-tuning stages, we further involve 1.2M SFT conversation data, as listed in Table 2 (including AI2D [33], DocVQA [11], and *etc*) to develop a high-resolution version, dubbed EVE-7B (HD).

## 4 Experiments

### 4.1 Training Settings

**Data Preparation.** (1) Pre-training Datasets. We train EVE using 33M publicly accessible samples from SA-1B [36], OpenImages [38], and LAION [63]. Following pre-processing pipelines [61, 46], we retain only samples with image resolutions higher than $448 \times 448$. In particular, the text data from LAION [63] are noisy and simplistic, lacking linguistic complexity. Besides, it suffers from duplication problems [78], which heavily hinder image diversity and quality. To extract more representative images, we apply K-means clustering technology on image features extracted by EVA-CLIP [71]. This process results in 50,000 clusters, from which we selected the 300 images closest to each cluster center, yielding a carefully curated subset of 15M image samples from LAION [63]. Subsequently, we utilize Emu2 (17B) and LLaVA-1.5 (13B) to regenerate high-quality image descriptions for the above three datasets and eliminate image-text samples with repetitive text or incomplete sentences. (2) Supervised Fine-tuning Datasets. We import LLaVA-mix-665K [49] for the standard EVE-7B, and collect a blended set of AI2D [33], Synthdog [34], DVQA [32], ChartQA [57], DocVQA [11], Vision-Flan [82], and Bunny-695K [27] for high-resolution version.

**Implementation Details.** EVE-7B is developed from Vicuna-7B [10] for vision-language domains. To control complexity, we limit the longest image edge to 672 for EVE-7B and 1344 for EVE-7B (HD), maintaining image aspect ratios unless otherwise specified. For visual supervision, we use CLIP-ViT-L/14 [61] as visual encoder and follow LLaVA-1.5 [49] protocols to add padding pixels and resize images to 336×336. Besides, the head is 8 for all cross-attention layers. The stride of the convolution layer and average pooling layer in PEL are 14 and 2, while the interval factor in PAL is 4. The patch features from EVE and VE are normalized by $\ell_2$-norm before MSE loss.

We adopt the AdamW optimizer [35], warm-up strategy, and cosine scheduler for training EVE. The maximum learning rates for Stage 1, 2, 3 are $4 \times 10^{-4}$, $4 \times 10^{-5}$, $2 \times 10^{-5}$, while the number of batch size and training samples are 512, 512, 128 and 16M, 33M, 665K for EVE-7B, which spends ~9 days using two 8-A100 (40G) nodes. Notably, we only implement a high-resolution training strategy in supervised fine-tuning stage to obtain EVE-7B (HD) by involving an extra 1.2M SFT data.

### 4.2 Main Results

We evaluate EVE on a series of public visual-language benchmarks, including academic-task-oriented benchmarks (VQA-v2 [25], GQA [29], VizWiz [26], and TextVQA [67]), hallucination benchmarks

Table 3: **Comparison with state-of-the-art VLMs on vision-language benchmarks.** #Samples: the number of samples in the pretraining/supervised fine-tuning stage. AR: image aspect ratio. HD: high image resolution. We evaluate VLMs on VQA$^{v2}$: VQA-v2 [25]; GQA [29]; VizWiz [26]; SQA$^I$: ScienceQA-IMG [54]; VQA$^T$: TextVQA [67]; POPE [47]; MME [23]; MMB: MMBench [52]; SEED: SEED-Bench [41]; MM-Vet [89]. $^\dagger$Includes in-house data that is not publicly accessible

| Method | LLM | #Samples | AR | VQA$^{v2}$ | GQA | VizWiz | SQA$^I$ | VQA$^T$ | POPE | MME | MMB | SEED | MM-Vet |
|---|---|---|---|---|---|---|---|---|---|---|---|---|---|
| *Encoder-based Vision-Language Models* | | | | | | | | | | | | | |
| InstructBLIP | Vicuna-7B | 129M/1.2M | Fix | – | 49.2 | 34.5 | 60.5 | 50.1 | – | – | 36.0 | 53.4 | 26.2 |
| IDEFICS-9B | LLaMA-7B | 353M/1M | Fix | 50.9 | 38.4 | 35.5 | – | 25.9 | – | – | 48.2 | – | – |
| QwenVL | Qwen-7B | 1.4B$^\dagger$/50M$^\dagger$ | Fix | 78.8 | 59.3 | 35.2 | 67.1 | 63.8 | – | – | 38.2 | 56.3 | – |
| QwenVL-Chat | Qwen-7B | 1.4B$^\dagger$/50M$^\dagger$ | Fix | 78.2 | 57.5 | 38.9 | 68.2 | 61.5 | – | 1487.5 | 60.6 | 58.2 | – |
| LLaVA-1.5 | Vicuna-7B | 558K/665K | Fix | 78.5 | 62.0 | 50.0 | 66.8 | 58.2 | 85.9 | 1510.7 | 64.3 | 58.6 | 30.5 |
| InternVL-Chat | Vicuna-7B | 4.98B/665K | Fix | 79.3 | 62.9 | 52.5 | – | 57.0 | 86.4 | 1525.1 | – | – | – |
| mPLUG-Owl2 | LLaMA2-7B | 400M/1.2M | Fix | 79.4 | 56.1 | 54.5 | 68.7 | 58.2 | 86.2 | 1450.2 | 64.5 | 57.8 | 36.5 |
| LVIS-4V | Vicuna-7B | 558K/885K | Fix | 79.6 | 62.6 | 51.8 | 68.3 | 58.7 | 86.0 | 1528.2 | 66.2 | 60.6 | 31.5 |
| ShareGPT4V | Vicuna-7B | 1.2M/665K | Fix | 80.6 | 63.3 | 57.2 | 68.4 | 60.4 | – | 1567.4 | 68.8 | – | 37.6 |
| Monkey (HD) | Qwen-7B | NA/1.44M | Enum | 80.3 | 60.7 | 61.2 | 69.4 | – | 67.6 | – | – | – | – |
| LLaVA-1.6 (HD) | Vicuna-7B | 558K/790K | Enum | 81.8 | 64.2 | 57.6 | 70.1 | 64.9 | 86.5 | 1519.3 | 67.4 | 64.7 | 43.9 |
| *Encoder-free Vision-Language Models* | | | | | | | | | | | | | |
| EVE-7B | Vicuna-7B | 33M/665K | Any | 75.4 | 60.8 | 41.8 | 63.0 | 51.9 | 83.6 | 1217.3 | 49.5 | 54.3 | 25.6 |
| Fuyu-8B (HD) | Persimmon-8B | –$^\dagger$/–$^\dagger$ | Any | 74.2 | – | – | – | – | 74.1 | 728.6 | 10.7 | – | 21.4 |
| EVE-7B (HD) | Vicuna-7B | 33M/1.8M | Any | 78.6 | 62.6 | 51.1 | 64.9 | 56.8 | 85.0 | 1305.7 | 52.3 | 56.8 | 25.7 |

(POPE [47]), open-world multi-modal understanding benchmarks (MME [23], MMBench [52], SEED-Bench [41], and MM-Vet [89]), scientific problem benchmarks (ScienceQA-IMG [54]).

(1) In Table 3, EVE demonstrates superior performance compared to Fuyu-8B [3], an encoder-free counterpart, across various vision-language benchmarks, despite its smaller size. The incorporation of diverse SFT datasets and larger image sizes in EVE (HD) significantly enhances its image recognition capabilities. This highlights how our adaptable image processing approach benefits from higher resolution inputs and more comprehensive instructional data. (2) EVE (HD) shows the competitive performance when compared to encoder-based VLMs, *e.g.*, InternVL-Chat [8], mPLUG-Owl2 [85], LVIS-4V [77], ShareGPT4V [7], and Monkey [48], without requiring additional visual encoders for complex image encoding. Notably, EVE (HD) outperforms several VLMs (InstructBLIP [12], IDEFICS-9B [30], QwenVL-Chat [2]), and performs on par with the well-regarded LLaVA-1.5 [49]. Notably, EVE faces challenges in accurately responding to specific instructions like option letters and binary questions. Besides, extensive training steps with only vision-language data have somewhat diminished its language competency and instruction-following capability. Consequently, the performance of EVE-7B on MME, MMB, SQA, and MM-Vet benchmarks is relatively subpar.

The fact that EVE-7B can match encoder-based VLMs using a simpler architecture and publicly available million-scale data is quite encouraging. This discovery suggests that encoder-free VLMs, in virtue of appropriate training strategies and high-quality image-text data, can efficiently achieve performance on par with or even surpass that of encoder-based VLMs. This opens up promising avenues for addressing the challenges typically associated with encoder-based VLMs, such as inflexible input modes, inefficient deployment, and inconsistent capacity across modalities.

### 4.3  Ablation Studies

**Configurations of patch embedding and aligning layer.** To validate the effectiveness of the proposed PEL and PAL, we conduct experiments with different configurations. $EVE_{0.5M}$, $EVE_{4M}$, and $EVE_{8M}$ represent models trained by LLaVA-pretrain-558K [49] (LLaVA-cap$_{558K}$), and subsets of 4M and 8M from the overall 33M EVE pre-training datasets (EVE-cap$_{4/33M}$, EVE-cap$_{8/33M}$) in Stages 1-2. All models use LLaVA-mix-665K [49] (LLaVA-mix$_{665K}$) in Stage 3. Besides, $MSE_{NP}$ means aligning features between the current EVE output and the next tokens from the vision encoder. From Table 4, (1) we observe that removing any cross-attention layer in PEL results in a marginal performance degradation; (2) Pairwise patch alignments perform slightly better than next-patch prediction, possibly because the vision-specific layer is shallow, causing the network to prefer the current patch encoding over predicting the next patch in an auto-regressive manner; (3) Whether using small-scale or large-scale training data, visual supervision via PAL effectively enhances fine-grained image representations, thereby facilitating visual perception and accelerating model convergence.

Table 4: Configurations of PEL and PAL. $EVE_{0.5M}$, $EVE_{4M}$, $EVE_{8M}$ utilize LLaVA-$cap_{558K}$, EVE-$cap_{4/33M}$, EVE-$cap_{8/33M}$ in Stage 1-2 with LLaVA-$mix_{665K}$ in Stage 3. $MSE_{NP}$ denotes the next-patch alignment

| Model | $VQA^{v2}$ | GQA | MMB | SEED |
|---|---|---|---|---|
| $EVE_{0.5M}$ | 58.5 | 49.8 | 35.3 | 39.3 |
| - PEL CA1 | 58.2 | 49.4 | 34.4 | 40.1 |
| - PEL CA2 | 58.4 | 49.8 | 35.8 | 39.3 |
| + $MSE_{NP}$ | 58.0 | 50.0 | 34.8 | 39.6 |
| - PAL | 55.6 | 47.5 | 34.5 | 37.8 |
| $EVE_{4M}$ | 69.4 | 56.5 | 42.0 | 48.7 |
| - PAL | 66.4 | 55.3 | 41.2 | 47.5 |
| $EVE_{8M}$ | 71.2 | 58.9 | 44.0 | 50.3 |
| - PAL | 69.4 | 57.3 | 42.7 | 49.2 |

Table 5: Configurations of training procedure. The top half all use LLaVA-$mix_{665K}$ for fine-tuning. The bottom half all use EVE-$cap_{33M}$ for pre-training

| Model | $VQA^{v2}$ | GQA | MMB | SEED |
|---|---|---|---|---|
| EVE w/o Stage 1 | | | | |
| + Stage 2 (4M) ↑ | 64.6 | 54.1 | 40.6 | 45.4 |
| + Stage 2 (8M) ↓ | 50.2 | 42.5 | 26.8 | 36.2 |
| EVE w/ Stage 1 | 59.3 | 51.0 | 36.5 | 39.7 |
| + Stage 2 (4M) ↑ | 69.4 | 56.5 | 42.0 | 48.7 |
| + Stage 2 (8M) ↑ | 71.2 | 58.9 | 44.0 | 50.3 |
| EVE w/ Stage 1-2 | | | | |
| + Stage 3 (665K) ↑ | 75.4 | 60.8 | 49.5 | 54.3 |
| + HD ↑ | 77.5 | 62.6 | 47.7 | 55.2 |
| + Stage 3 (1.8M) ↑ | 76.7 | 61.8 | 51.2 | 54.8 |
| + HD ↑ | 78.6 | 62.6 | 52.3 | 56.8 |

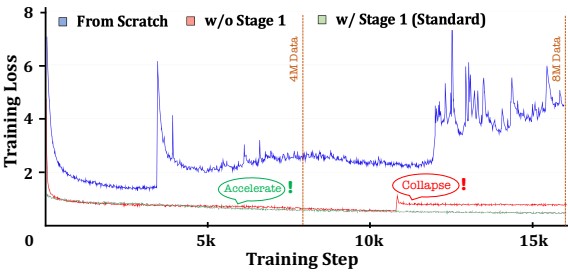

Figure 5: Training loss in Stage 2 using various strategies. Optimization remains unstable and prone to collapse, despite searching learning rate from 2e-5 to 1e-3.

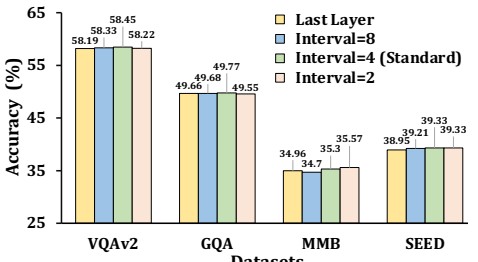

Figure 6: Configurations of interval factor. We observe that only the last vision output is insufficient and interval = 4 works the best.

**Stage 1: LLM-guided pre-aligning stage does count for efficient training.** In Table 5, we display the influence of our training strategies, including pre-training and supervised fine-tuning stages. We discover that (1) without LLM-guided pre-aligning between vision and language in Stage 1, the entire pre-training process becomes unstable and suffers from sudden collapse. This results in an unusual situation where performance rapidly decreases as data volume increases beyond a certain point, as shown in Figure 5. We assume that the randomly initialized PEL introduces significant noise and negative interference into network optimization for severely turning LLM into VLM, exacerbated by accumulation problems. Hence, Stage 1 stabilizes the training process and seamlessly transitions into Stage 2, emphasizing its indispensable role in efficient training to construct an encoder-free VLM. This stage prevents collapses and expedites convergence, guaranteeing a smooth and robust training route. (2) Increasing image resolutions and incorporating diverse instruction data in Stage 3 bring extreme performance gains after obtaining a well-aligned state across modalities in Stage 1-2. The high-resolution property greatly enhances fine-grained visual recognition, particularly for charts, tables, OCR information, and high-definition samples. Moreover, due to the extensive pre-training data, there is a pressing need for more supervised fine-tuning data to improve inherent linguistic proficiency, commonsense knowledge, and instruction-following patterns forgotten by EVE. (3) Training VLMs from scratch in Figure 5 often leads to instability and poses significant challenges for model optimization. Therefore, LLMs serve as an effective initialization for developing VLMs.

**Merging patch features across layers outperforms the last layer output.** Figure 6 illustrates the impact of varying interval factors for selecting cross-layer features in PAL. Optimal performance emerges when the interval factor is set to 4, yielding peak scores across VQA-v2, GQA, and SEED benchmarks. Our findings solidify the superiority of multi-layer aggregation over singularly relying on the last layer output. That is because a competitive interplay unfolds within the dynamic mapping space between vision and text, resulting in projection coupling and mutual interference within the last layer of LLM. Furthermore, hierarchical integration for vision features empowers multi-level representations, facilitating adaptive alignments with abundant characteristics from the vision encoder.

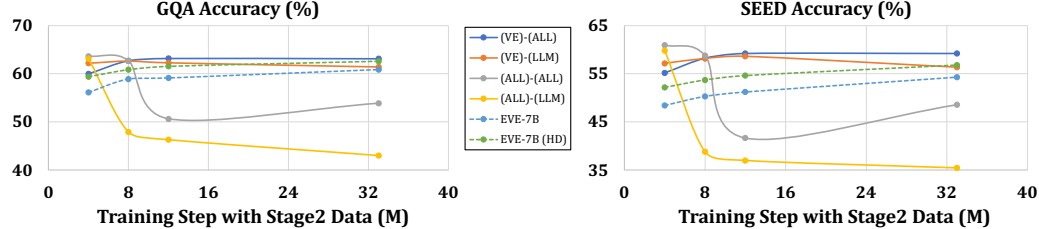

Figure 7: Scaling performance on GQA and SEED using LLaVA-1.5 as an encoder-based baseline. We first train its projector in Stage 1 with EVE-cap$_{16/33M}$. Here, (VE)-(LLM) indicates training Vision Encoder in Stage 2 and LLM in Stage 3, where we train the projector across all stages.

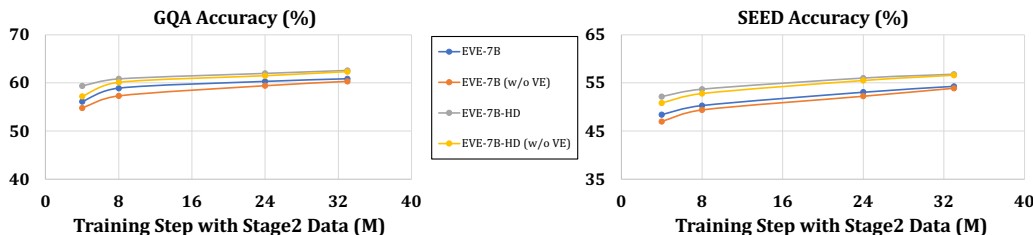

Figure 8: Scaling performance on GQA and SEED w/ or w/o vision encoder supervision (PAL).

**EVE consistently improves with more pre-training data, nearing encoder-based performance.** In Figure 7, we first train LLaVA-1.5's projector via EVE-cap$_{16/33M}$ in Stage 1, and then attempt various training strategies in Stage 2-3 following [85, 2, 8, 9, 53, 7]. For example, (VE)-(ALL) denotes training vision encoder in Stage 2 and all model parameters in Stage 3. We discover that: (1) Using synthetic captions, encoder-based model often suffers from collapse [88] due to simplified language structure and lack of knowledge details. Only the (VE)-(ALL) strategy avoids this issue and outperforms the baseline by freezing LLM weights during pre-training to preserve language proficiency and world knowledge. (2) EVE shows consistent performance gains with scaled-up data, gradually approaching the encoder-based model's performance. This may be because encoding and aligning vision and text in a unified network is more challenging and requires more training data, making encoder-free models less prone to overfitting compared to encoder-based ones. (3) For the vision encoder (CLIP) within the PAL of EVE, we discover that unfreezing it at all stages results in model collapse, which does not yield any improvements and instead increases computational costs.

**Vision supervision aids in early convergence but becomes less critical during large data scale-up.** Vision encoder supervision improves training efficiency, especially with a relatively limited data scale compared to the billion scale of CLIP image-text pairs. Actually, this component can be simplified with sufficient data resources as shown in Figure 8. Our findings indicate that its influence diminishes over large data scales, and by the 24M mark, the difference in performance with or without this supervision is about 0.3-0.8%. This may be because large amounts of high-quality and detailed captions greatly enhance the understanding of visual information, thus gradually reducing the need for visual supervision. We emphasize that vision supervision (PAL) is not the crucial factor for training stability and scaling efficiency. Even with vision supervision, performance without LLM-guided pre-aligning in Stage 1 rapidly decreases as data volume increases beyond a certain point.

**Efficient deployment and lower latency.** Table 6 demonstrates several inference metrics, including floating point operations per second (FLOPs) and time delay. By eliminating the deep pre-trained vision encoder, EVE significantly accelerates the image encoding process, achieving an order of magnitude speed improvement over its counterparts. The CA1 component in PEL serves to control computational complexity. Even with a minimal downsampling rate of 2, EVE (HD) surpasses LLaVA-1.6 in both efficient deployment and reduced inference

Table 6: Model FLOPs and inference latency

| Model | Vision Part | | LLM Part | |
| | FLOPs(G) | Time(s) | FLOPs(T) | Time(s) |
| --- | --- | --- | --- | --- |
| LLaVA-1.5 | 372 | 0.033 | 15.2 | 0.48 |
| EVE-7B | 42 | 0.003 | 15.2 | 0.48 |
| LLaVA-1.6 (HD) | 1860 | 0.13 | 76.1 | 2.07 |
| EVE-7B (HD) | 170 | 0.013 | 60.8 | 1.52 |

delay. By optimizing FLOPs and minimizing inference latency, EVE ensures rapid and efficient processing. This leads to a more responsive and resource-efficient system, positioning encoder-free VLMs as superior to encoder-based VLMs in practical applications.

## 5 Limitation and Discussion

EVE demonstrates a desired and powerful encoder-free decoder-only architecture that essentially solves various issues of existing encoder-based VLMs. Despite its promising results, a key limitation is the performance gap that still exists between EVE and state-of-the-art encoder-based VLMs. Additionally, due to time constraints, several questions and considerations remain:

**(1) Further performance gain.** We empirically find that training solely with vision-language data significantly reduces language capability (SQA score drops from 65.3% to 63.0%) while gradually improving multimodal performance. This indicates catastrophic forgetting of linguistic knowledge within the LLM. To address this, we suggest merging language and multimodal data appropriately or employing mixture-of-experts (MoE) strategies to mitigate vision-language interference.

**(2) Encoder-free expectation.** We validate that encoder-free VLMs can rival encoder-based VLMs but require larger training samples. How about the performance under the same LLM capacity and training data? We are working on the assumption that scaling up LLM capacity and training data would enable encoder-free VLMs to match or even surpass encoder-based VLMs, due to the nearly lossless image inputs that bypass pre-processing by vision encoders.

**(3) Multi-modality inspiration.** As a bonus, we gain inspiration for integrating additional modalities (*e.g.*, audio, video, thermal, depth, *etc*.). The core idea is to pre-align these modalities with a frozen LLM from an LLM-centric perspective before introducing large-scale unified training, supervised by corresponding single-modality encoders and linguistic conceptual alignments.

## 6 Conclusion

In this paper, we introduce EVE, a simple yet effective encoder-free model with a decoder-only architecture, designed to revolutionize vision-language understanding. Rather than merely presenting another VLM, we aim to unveil the under-explored bottlenecks towards constructing an encoder-free VLM. Our key findings reveal that: (1) An LLM-centric perspective, where vision and language modalities are pre-aligned, delivers robust, efficient, and consistent improvements as the pre-trained data scale up. (2) Incorporating fine-grained alignment with pre-trained vision encoders and linguistic conceptual supervision significantly enhances vision recognition, resulting in faster convergence and substantial performance gains. Remarkably, EVE performs on par with mainstream encoder-based VLMs solely relying on 35M publicly available data, and dramatically outperforms the counterpart Fuyu-8B, trained by undisclosed strategies and private datasets. As a pioneer, EVE provides a transparent and efficient roadmap for future VLM development, essentially tackling the challenges of multi-modal input processing, deployment efficiency, and model capacity across modalities. In the future, we are devoted to scaling up the model capacity with high-quality training data to explore the limits of encoder-free VLMs. Besides, we are working on translating the single-modality LLMs to large multimodal models with more modalities through our proposed training strategies.

## Acknowledgments

This project is supported by the National Key R&D Program of China (2022ZD0116302), the National Natural Science Foundation of China under grant No. 62293540 and 62293542, Liao Ning Province Science and Technology Plan No.2023JH26/10200016, and Dalian City Science and Technology Innovation Fund No. 2023JJ11CG001.

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

# A Appendix

## A.1 Ethics Statement

We introduce EVE, a pioneering encoder-free VLM, along with efficient training strategies for developing an encoder-free VLM. It is important to address the ethical considerations surrounding large pre-trained models, such as potential bias or discrimination [6] and privacy concerns [4] in the extensive training data. Additionally, the computational cost and environmental impact have become significant topics of discussion [69]. To our knowledge, previous VLMs have not thoroughly investigated their impact on issues like bias or information leakage, which warrants further exploration. Unlike closed-source VLMs, we train EVE using publicly available data, thereby raising fewer ethical concerns. Furthermore, our approach reduces model deployment costs by lowering computational consumption and server resource requirements. Notably, our training experiments are conducted in a data center powered entirely by renewable energy sources.

## A.2 Dataset Details

In Table 7, we provide a detailed description of SFT datasets, along with their sampling rates during Stage 3. The majority are derived from LLaVA-mix-665K [49] and Bunny-695K [27]. Besides, we compile a blended set from AI2D [33], Synthdog [34], DVQA [32], ChartQA [57], DocVQA [11] and Vision-Flan [82] to leverage the advantages of high-resolution inputs.

Table 7: **Supervised Fine-tuning Data Mixture**

| Data | Sample Raito | Size | Task |
|---|---|---|---|
| LLaVA [49] | 8.7% | 158K | Conversation |
| SVIT-core-150K [90] | 8.7% | 158K | |
| ShareGPT [7] | 2.2% | 40K | Text-only |
| WizardLM-70K [80] | 3.9% | 70K | |
| VQAv2 [25] | 9.1% | 83K | General QA |
| GQA [29] | 7.9% | 72K | |
| OKVQA [56] | 1.0% | 9K | |
| A-OKVQA [64] | 7.3% | 66K | Knowledge QA |
| TextCaps [66] | 2.4% | 22K | OCR QA |
| Synthdog [34] | 1.7% | 30K | |
| OCRVQA [59] | 8.8% | 80K | |
| RefCOCO [87, 55] | 5.3% | 48K | Grounding |
| VG [37] | 9.5% | 86K | |
| Vision-Flan [82] | 10.2% | 186K | Multi-task |
| ChartQA [57] | 1.0% | 18K | Chart QA |
| DVQA [32] | 11.0% | 200K | |
| AI2D [33] | 0.7% | 12K | Science QA |
| DocQA [11] | 0.6% | 10K | Document |
| Total | 100.0% | 1.35M | |

Table 8: **Hyperparameter settings**

| Hyperparameter | Stage-1 | Stage-2 | Stage-3 |
|---|---|---|---|
| Batch Size | 512 | 512 | 128 |
| Learning Rate (lr) | 4e-4 | 4e-5 | 2e-5 |
| LR Schedule | | cosine decay | |
| LR Warmup Ratio | 0.03 | 0.01 | 0.01 |
| Weight Decay | | 0 | |
| Epoch | | 1 | |
| Optimizer | | AdamW | |
| DeepSpeed stage | 3 | 3 | 3 |

## A.3 Training Details

We report the detailed training hyper-parameter settings of EVE during the three stages in Table 8.

## A.4 Qualitative Results

The qualitative results are demonstrated for comparisons. We showcase the ability of EVE in the aspects of OCR, reasoning, and image captioning, respectively.

| | |
|---|---|
| **Visual input example** | |

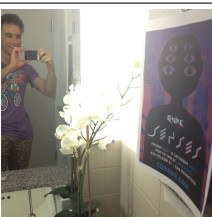

| | |
|---|---|
| User | Please describe image contents in detail. |
| LLaVA-1.5 | The image features a man with an unusual appearance, standing in front of a mirror. He is wearing a purple shirt and appears to be a robot or a character from a comic. The man is taking a selfie with his cell phone, holding it up to his face while capturing the moment. In the scene, a potted plant is positioned nearby, adding a touch of greenery to the setting. Two other people can be seen in the background, one situated to the left and another towards the right, but they are not the main focus of the scene. Additionally, a bottle is visible at the far left of the image, potentially a decorative item or part of the room's design. |
| EVE | The image features a man taking a picture of himself in a bathroom mirror. He is standing in front of a sink, which is located on the left side of the room. The man is holding a cell phone in his hand, capturing the moment. The bathroom is decorated with a few potted plants, one of which is placed near the sink, and another one is located closer to the right side of the room. There is also a vase in the scene, positioned near the center of the room. |
| EVE (HD) | The image depicts a bathroom scene. A man is taking a selfie in the mirror, which is mounted on the wall. He is wearing a purple shirt and leopard print pants. He is holding a camera, capturing his reflection. The mirror is large and spans a significant portion of the wall. The bathroom features a white sink with a silver faucet, which is located on the left side of the image. The sink is positioned on a countertop. Above the sink, there is a mirror. The countertop is made of granite and has a white tile backsplash. To the right of the sink, there is a white tiled wall. A white tiled floor is visible in the lower part of the image. The bathroom also contains a plant with white flowers, which is placed in a vase. The plant is located near the sink and the mirror. The flowers are white and are arranged in a bouquet. There is also a poster on the wall, which is positioned to the right of the mirror. The poster is black and white and is quite large. It features a picture of a remote control and some text. The image also includes a white tile wall, which is located behind the man. The wall is clean and well-maintained. Overall, the image captures a moment in a bathroom, with a man taking a selfie in a mirror, a plant with white flowers, a sink, a mirror, a poster, and a tiled wall. |

| | |
|---|---|
| **Visual input example** | |

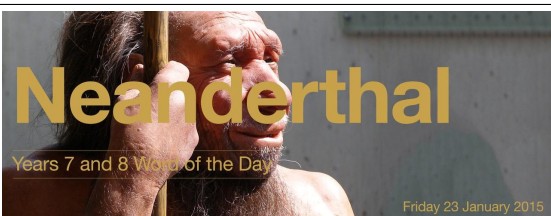

| | |
|---|---|
| User | Please describe image contents in detail. |
| LLaVA-1.5 | The image depicts an unconventional scene of a man wearing a costume with a jaw, nose, and chin made of rocks. This unique character appears to be a humanoid figure made from stones, with a gold background that highlights the costume. The man's face, with its interesting rock-formed facial features, seems to be the focal point of the image. |
| EVE | The image features a man holding a wooden stick, possibly a staff or a walking stick, in front of a building. The man appears to be an older individual with a beard. He is standing in front of a wall, which could be a part of the building or a separate structure. The scene is set outdoors, with the man being the main focus of the image. |
| EVE (HD) | The image features a man with a long beard and a bald head. He is holding a wooden stick in his hand. The man's face is partially covered by a large yellow text that reads "Neanderthal". The man's face is also partially covered by a large yellow text that reads "Years 7 and 8". The man's face is also partially covered by a large yellow text that reads "Worst of the Day". The man's face is also partially covered by a large yellow text that reads "Friday 23 January 2015". The man's face is also partially covered by a large yellow text that reads "Years 7 and 8". The man's face is also partially covered by a large yellow text that reads "Neanderthal". |

