# OpenReview forum: "Unveiling Encoder-Free Vision-Language Models"
_NeurIPS.cc/2024/Conference — NeurIPS 2024 spotlight_

### Official Review · Reviewer_Z41h · 2024-07-02

**Soundness:** 4
**Presentation:** 4
**Contribution:** 3
**Rating:** 6
**Confidence:** 5

**Summary:**

This paper introduces EVE, a novel paradigm for Vision-Language Models (VLMs) designed to eliminate the need for a preceding visual encoder in the LLM decoder. This approach aims to create a more flexible and lightweight vision-language framework. EVE incorporates a Patch Embedding Layer and a Patch Aligning Layer to align image tokens with the language model. Additionally, it proposes a new training recipe for robust encoder-free model learning. The result is an efficient VLM that achieves performance (somewhat) comparable to existing encoder-based VLMs across several benchmarks.

**Strengths:**

[**Writing**] The paper exhibits exceptional clarity and coherence in presenting its concepts. The accompanying figures serve as effective visual aids, significantly enhancing the comprehensibility of the discussed ideas.

[**Motivation**] The rationale behind removing off-the-shelf encoders in current VLMs is compelling and innovative. While most existing works adhere to the well-established encoder-based VLM paradigm, this exploration into a new direction is commendable. Such investigations are invaluable to the research community.

[**Methodology Transparency**] The authors provide comprehensive details on dataset curation and implementation specifics.

[**Ablation Studies**] The inclusion of detailed ablation experiments is a significant strength of this paper. These studies collectively offer a clear demonstration of the proposed model's capabilities and distinctive features, providing crucial insights into the model's performance and design choices.

**Weaknesses:**

[**Encoder-Free?**] The model's positioning as "encoder-free" is somewhat misleading, given the presence of a Patch Embedding Layer (PEL) that functions very similarly to an image encoder. Despite its light-weight and compact nature, the PEL's architecture closely resembles a ViT encoder. Furthermore, the proposed Patch Aligning Layer (PAL) encourages CLIP-like token generation, further blurring the line between this approach and traditional encoder-based models.

[**Almost Lossless?**] The manuscript repeatedly claims that the PEL converts images into embeddings "almost losslessly." However, the use of average pooling for feature map downsampling is inherently a lossy operation, undermining this claim.

[**Performance Gap**] The model's performance, while promising, still lags behind encoder-based VLMs. It's notable that models like LVIS-4V, ShareGPT4V, and LLaVA-1.6 achieve superior results despite utilizing substantially less training data than EVE.

[**Complexity Analysis**] Table 6 demonstrates significant reductions in FLOPs and inference time for the visual encoding procedure. However, this improvement should be contextualized within the overall model architecture, where the LLM block dominates computational demands. This raises questions about the relative impact of optimizing the visual encoder in terms of overall efficiency gains.

[**Flexibility Benefits**] The manuscript emphasizes the advantages of supporting flexible Image Resolution (IR) and Aspect Ratio (AR) as key benefits over encoder-based VLMs. However, the practical significance and impact of this flexibility are not comprehensively demonstrated or evaluated or quantified in the current study.

[**Structural Issues**]
* Figure 1 is never referenced in the main text.
* Figure 5 is mentioned after Figure 6 in the text, disrupting the logical flow.

**Questions:**

I love the concept and the exploration of encoder-free VLMs. However, as mentioned in the weaknesses, several points significantly undermine the significance of this work and the key claims of it. Here are several important questions.
1. Given that the Patch Embedding Layer (PEL) functions similarly to an image encoder, how do you justify classifying EVE as an "encoder-free" model?
2. The manuscript claims that the PEL converts images into embeddings "almost losslessly." How do you elaborate this claim with the use of average pooling, which is inherently a lossy operation?
3. Despite using more training data, EVE's performance is still lower than some encoder-based VLMs like LVIS-4V, ShareGPT4V, and LLaVA-1.6. What factors do you believe contribute to this performance gap, and what strategies are you considering to address it?
4. While Table 6 shows significant reductions in FLOPs and inference time for visual encoding, the LLM block remains the computational bottleneck. How significant do you consider the efficiency gains from the encoder-free approach in the context of the overall model performance?
5. The paper emphasizes the advantages of flexible Image Resolution (IR) and Aspect Ratio (AR) support. Could you provide specific examples or use cases where this flexibility offers significant practical benefits over traditional encoder-based models?

**Limitations:**

The discussion on limitations and societal impact about the paper is relatively thorough. The points raised in the weaknesses section further articulate limitations that should be considered.

---

> ### Author Rebuttal · Authors · 2024-08-07
>
> We are grateful for your meticulous and insightful review. We have carefully considered your questions and polished our paper.
>
> `Q1: [Encoder-Free?] How do you justify classifying EVE as an "encoder-free" model?`
>
> **(1) A more essential difference between PEL and image encoders is whether they involve strong inductive bias and semantic priors in abstracting visual representation by pre-trained models via pretext tasks.** We introduce CA1 and CA2 mainly due to efficiency consideration and holistic information for raster-order token inputs respectively. Note that Table 4 shows that CA1 and CA2 are optional and have a less significant impact on performance.
>
> **(2) As discussed in ALL-Q1, we found that while PAL aids in early convergence and training efficiency, it becomes less crucial as the data scale significantly increases.**
> Besides, we can completely remove it during inference, allowing EVE to function as a pure encoder-free architecture like Fuyu-8B. Hence, we attribute our work to the ‘encoder-free’ track.
>
> `Q2: [Almost Lossless?] The use of average pooling, which is inherently a lossy operation?`
>
> Good question. CA1 may not deviate from the "almost" lossless concept due to its dense feature aggregation strategy in a restricted receptive field, only using pooling features as the query. We adopt this design, mainly considering the trade-off between large token numbers and training efficiency, especially with limited resources.
>
> `Q3: [Performance Gap?] [Complexity Analysis?] [Flexibility Benefits?]`
>
> **(1) We highlight that EVE shows potentially more promising scaling with pretraining duration, which is the key motivation behind our efforts to build encoder-free VLMs.**
> EVE attempts to remove strong vision inductive bias and transmit visual signals almost losslessly for better scaling properties.
> In Figure 2 of the rebuttal PDF, we observe that encoder-based models often suffer from collapse. Only the (VE)-(ALL) training strategy avoids this issue by freezing LLM weights during pre-training and unfreezing them during the SFT stage. In contrast, EVE shows better scaling properties and gradually approaches the performance of well-developed encoder-based VLMs with only 33M data scale.
>
> **(2) Encoder-free VLMs are promising for scaling but require enormous training data to develop vision perception from scratch.**
> Here, with only 33M pre-training data, our pioneering exploration currently lags behind but performs comparably to popular encoder-based methods. **Interestingly, subsequent PaliGemma [a]** also explores an encoder-free version via 1B pre-training image-text pairs, showing promising early results alongside its encoder-based counterpart across 37 validation datasets (see Figure 3 of the rebuttal PDF). They particularly mention that only the separate vision encoder i.e. SigLIP, has been trained with 40B image-text pairs, far greater than 1B data of encoder-free version. They also indicate that decoder-only VLMs may be a promising direction although currently suffering in training efficiency due to building vision perception from scratch.
>
> **(3) The 'Any' image ratio, simple architecture, and efficient deployment are bonuses of encoder-free VLMs.**
> Recent studies on encoder-based VLMs reveal that
> **(i)** Due to the limitations of pre-trained encoders, existing VLMs exhibit vulnerabilities in basic capabilities rooted in visual encoding trade-off [b, c].
> **(ii)** Various vision encoders show uneven levels of capability due to pretext pretraining tasks, relying heavily on the corrective capabilities of LLMs for multimodal understanding [d, e].
> In contrast, encoder-free VLMs remove semantic priors in abstracting visual representation, theoretically allowing VLMs to autonomously acquire all available information. **While 'any image ratio' and 'FLOPS gains' are natural benefits of the encoder-free approach, the primary reason for exploring an encoder-free model is its scaling efficiency with less inductive bias.**
> In this premise, removing the vision encoder provides only a modest bonus in terms of flexible image input and deployment efficiency. Notably, the encoder-free track is still in early development and has a long way to explore its limits.
>
> [a] PaliGemma: A versatile 3B VLM for transfer. Google DeepMind. arXiv 2407.07726.
>
> [b] LLaVA-UHD: an LMM Perceiving Any Aspect Ratio and High-Resolution Images. Xu et al. arXiv 2403.11703.
>
> [c] HiRes-LLaVA: Restoring Fragmentation Input in High-Resolution Large Vision-Language Models. arXiv 2407.08706.
>
> [d] Eyes Wide Shut? Exploring the Visual Shortcomings of Multimodal LLMs. Tong et al. CVPR2024.
>
> [e] Cambrian-1: A Fully Open, Vision-Centric Exploration of Multimodal LLMs. Tong et al. arXiv 2406.16860.
>
> `Q4: The performance gap with some encoder-based VLMs? How to address it?`
>
> **(1) The key issue with EVE is the massive data needed to build vision perception from scratch.**
> Encoder-based models first train a separate visual encoder with billions of data points, while EVE constructs vision perception from scratch using only 33 million data points. Figures 2-3 show performance is far from saturated. Larger data scales and more computing resources are required for further exploration.
>
> **(2) Besides, several important factors deserve to be explored**. We found image resolution, reducing modality perturbations, the base LLM’s size and capability, data size and quality, etc. also impact the scaling efficiency of encoder-free VLMs. For instance, pre-training EVE with 4M high-resolution data (1344 longest edge) without vision supervision, can match the performance of the original version, using 12M data and vision supervision. We would polish these factors and develop a fully and stronger encoder-free VLM in the follow-up study.
>
> `Q5: Figure 1 is never referenced in the main text. Figure 5 is mentioned after Figure 6 in the text, disrupting the logical flow.`
>
> Thanks for your kind reminder and we will polish them in the revised manuscript.

---

> > ### Comment · Reviewer_Z41h · 2024-08-08
> > **Thank the authors for the response**
> >
> > Thank the authors for your response. The rebuttal clears most of my previous concerns. On a side note, I still do not like the term "encoder-free" and "lossless". In my opinion, it is more accurate to say that this work explores joint training of the LLM and a light-weight vision encoder from scratch. However, I do agree with other points made my the authors in the rebuttal. Hence, I have decided to keep my positive rating.

---

> > > ### Author Response · Authors · 2024-08-11
> > > **New reply and clarification to reviewer Z41h**
> > >
> > > We are pleased to reach a consensus on key insights of EVE. We understand your concerns regarding the terms "encoder-free" and "lossless" description, and we would like to clarify our rationale behind these choices:
> > >
> > > `Q1: On a side note, I still do not like the term "encoder-free" and "lossless". In my opinion, it is more accurate to say that this work explores joint training of the LLM and a light-weight vision encoder from scratch.`
> > >
> > > **(1) While there's nothing wrong with regarding EVE as joint training of the LLM and a light-weight vision layer, we intentionally adopted the terms "encoder-free" and "lossless" to highlight the different design principles that distinguish EVE from existing modular VLMs.**
> > >
> > > Actually, we have explored the simplest setup. By removing the PAL and introducing the PEL via a streamlined Conv1 (14×14×1024)-GeLU-Conv2 (2×2×4096) structure, we ensured that the total number of activation values remains unchanged. This alignment with the "encoder-free" and "lossless" concepts reflects our goal to chase maximum simplicity without the need for a conventional, standalone vision encoder. Our findings were as follows:
> > >
> > > - The LLM-guided Prealigning Stage effectively prevents model collapse, which is a challenging issue in building encoder-free VLMs.
> > >
> > > - The minimalist PEL and drop of PAL initially impact training efficiency, particularly during early convergence (about 2.1-4.4\% under 5M high-resolution pretraining data). However, this gap gradually decreases as the data scale increases. This behavior underscores the LLM's great capability to learn visual perception and establish multimodal alignment from scratch, even in the absence of a vision encoder.
> > >
> > > The current design, although a trade-off, was chosen to prove the feasibility of an encoder-free VLM that can rival encoder-based ones, especially given the previously limited data and device resources.
> > >
> > > **(2) Once again, we would like to stress that the core idea behind EVE is to challenge the prevailing inductive biases present in encoder-based VLMs by allowing the model maximum freedom to discover more rational patterns on its own, inspired by 'The Bitter Lesson' (Rich Sutton).** This design philosophy aligns with the new trend in VLM research (GPT-4o [a], "discrete" Chameleon [b] and MoMa [c], 'continuous' Fuyu-8B [d] and SOLO [e]), where the focus is on reducing architectural constraints to develop an end-to-end VLM and process textual and visual inputs through a unified network for better scaling efficiency.
> > >
> > > **(3) We have preliminarily demonstrated that EVE is not only feasible but also promising for future advancements in VLMs. This innovation is what we believe sets EVE apart and warrants a re-evaluation of our proposed terminology.**
> > >
> > > We respectfully hope that this clarification will prompt you to reconsider your assessment and potentially raise the score of our submission. Should you have any further questions or require additional details, we would be happy to provide them.
> > >
> > > [a] GPT-4o System Card. OpenAI. August 8, 2024.
> > >
> > > [b] Chameleon: Mixed-Modal Early-Fusion Foundation Models. Meta FAIR. arXiv 2405.09818.
> > >
> > > [c] MoMa: Efficient Early-Fusion Pre-training with Mixture of Modality-Aware Experts. Meta FAIR. arXiv 2407.21770.
> > >
> > > [d] Fuyu-8B: A Multimodal Architecture for AI Agents. Adept AI. October 17, 2023.
> > >
> > > [e] SOLO: A Single Transformer for Scalable Vision-Language Modeling. UIUC. arXiv 2407.06438.

---

### Official Review · Reviewer_nP6r · 2024-07-09

**Soundness:** 3
**Presentation:** 3
**Contribution:** 3
**Rating:** 6
**Confidence:** 3

**Summary:**

This paper revisits the vision-encoder-free MLLM direction, which is not a popular choice in the community at present. A new method EVE is proposed to reduce the gap between encoder-free and encoder-based MLLMs, demonstrating a large improvement against Fuyu-8B (the best encoder-free MLLM so far). EVE introduce additional modules to distill visual features from an existing vision encoder, then align image-text in the consequential stage.

**Strengths:**

The recipe (EVE) shared by this paper is simple yet effective, which could be insightful to the VLM community to develop better encoder-free MLLMs, which have many benefits, such as flexible resolution and aspect ratio, efficient deployment and lower latency etc.

The experiments are comprehensive, with a large number of baselines.

The paper is well written and easy to read.

**Weaknesses:**

IIUC, Fuyu learns visual features from scratch, while EVE actually distillates from  an existing CLIP-ViT-L-336px vision encoder, so it’s not fair to claim that EVE is better than Fuyu under the same “encoder free” setting. It’s more like “encoder distilled” vs. “encoder free”. Therefore, it would be more insightful if we could ablate encoder free vs. distilled in 4.3 (if the training stability issue can be solved), i.e., to investigate why EVE is much better than Fuyu.

As mentioned in the paper, the language ability is largely affected in stage 2. It’s very common to add an additional LM task and there have been many open-sourced corpuses already. It’s worth explaining why this common approach was not tried (though the paper suggests to solve this in future work).

Even though EVE-7B is comparable with the selected encoder-based baseline models of the same size, it is still largely behind some recent models with even half size and lower resolution, such as PaliGemma-3B [1].

[1] https://ai.google.dev/gemma/docs/paligemma/model-card

**Questions:**

It’s unclear whether 35m data is enough (likely not?). Have you tried data scaling ablations?

In line 234, it should be Table 3?

**Limitations:**

It’s worth mentioning EVE inherits the same limitations from the resued LLM, i.e., Vicuna-7B (likely also including CLIP-ViT-L-336px).

---

> ### Author Rebuttal · Authors · 2024-08-07
>
> Your constructive comments are much appreciated. We have addressed all your points and revised the paper accordingly to ensure its improvement.
>
> `Q1: IIUC, Fuyu learns visual features from scratch, while EVE actually distillates from an existing CLIP-ViT-L-336px vision encoder, so it’s not fair to claim that EVE is better than Fuyu under the same “encoder free” setting. It’s more like “encoder distilled” vs. “encoder free”. Therefore, it would be more insightful if we could ablate encoder free vs. distilled in 4.3 (if the training stability issue can be solved), i.e., to investigate why EVE is much better than Fuyu.`
>
> **(1) As discussed in ALL-Q1, while the vision encoder aids in early convergence, it becomes less crucial as the data scale increases.**
> We introduce vision encoder distillation to improve training efficiency, especially with a limited data scale, which is less significant after 24M pre-training data in our experiments. This indicates that eliminating the vision encoder for a fully encoder-free VLM is practical during both training and inference.
>
> **(2) The LLM-guided pre-aligning stage is one crucial factor for training stability and scaling efficiency.**
> Without this stage, even with enormous training data and vision distillation, training decoder-only VLMs remains highly challenging in Table 5 and Figure 6, suffering from model collapse and large performance gaps.
>
> **(3) Compared with Fuyu-8B, EVE adopts image resolution with 672 image longest edge during pre-training due to device constraints and efficiency consideration.**
> For fairness, we attempt to pre-train EVE with 4M high-resolution data (1344 longest edge) without vision supervision, which can match the performance of our original version, using 12M data and vision supervision. This indicates that EVE has room for further improvement and could potentially widen the gap beyond Fuyu-8B.
>
> `Q2: As mentioned in the paper, the language ability is largely affected in stage 2. It’s very common to add an additional LM task and there have been many open-sourced corpuses already. It’s worth explaining why this common approach was not tried (though the paper suggests solving this in future work).`
>
> While maintaining language ability in stage 2 is a concern, it is not the primary challenge in developing encoder-free VLMs effectively. The main issue is addressing optimization problems and bridging the gap in multi-modality capabilities compared to encoder-based VLMs. We spent a long time exploring and finally discovered that an LLM-guided pre-aligning stage and vision encoder supervision are essential for training stability and early convergence, respectively. Considering text-only data is less critical at this stage, we decided to solve it in the follow-up study.
>
> `Q3: Even though EVE-7B is comparable with the selected encoder-based baseline models of the same size, it is still largely behind some recent models with even half size and lower resolution, such as PaliGemma-3B [1].`
>
> **(1) Actually, such a comparison is unfair due to many notable differences.**
> - [Data Magnitude]: As mentioned before, EVE learns vision perception from scratch, where the vision encoder could be removed using the entire 33M pretraining data. However, PaliGemma’s SigLIP encoder has seen 40B image-text pairs during Stage 0, with Stage 1 and Stage 2 further processing about 350B and 90B tokens, respectively.
> - [LLM Capability]: Despite being smaller, Gemma-2B exhibits similar capabilities to our Vicuna-7B.
> - [Finetuning Stategy]: Table 1 in PaliGemma shows that they train a 'specialist' model for each task with specific hyperparameters, while EVE serves as a 'generalist' model for downstream tasks.
>
> **(2) We highlight that EVE shows potentially more promising scaling with pretraining duration.**
>
> In Figure 2 of the rebuttal PDF, we observe that EVE shows better scaling properties and gradually approaches the performance of well-developed encoder-based VLMs with only 33M data scale. **Interestingly, PaliGemma** also explores an encoder-free version via 1B pre-training image-text pairs, showing promising early results alongside its encoder-based counterpart across 37 validation datasets (see Figure 3 of the rebuttal PDF). They particularly mention that their vision encoder i.e. SigLIP, was trained with 40B image-text pairs, and indicate that decoder-only VLMs may be a promising direction although currently suffering in training efficiency due to building vision perception from scratch.
>
> `Q4: It’s unclear whether 35m data is enough (likely not?). Have you tried data scaling ablations?`
>
> 35M data points provide valuable insights for constructing encoder-free VLMs but fall short of peak performance. Figure 1-2 in the rebuttal PDF show that performance is not yet saturated, and EVE offers better training stability and scaling efficiency. Due to budget and time constraints, we will scale the pre-training corpus to a billion scale in future work.
>
> `Q5: In line 234, it should be Table 3?`
>
> Thanks for the kind reminder. We will revise it.
>
> `Q6: It’s worth mentioning EVE inherits the same limitations from the resued LLM, i.e., Vicuna-7B (likely also including CLIP-ViT-L-336px).`
>
> This is a common problem for both encoder-based and encoder-free VLMs. We believe that as existing LLMs continue to advance impressively, these limitations, we think, would be properly addressed. Besides, encoder-free VLMs escape the limitations of pre-trained vision encoders, unlike their encoder-based counterparts.

---

> > ### Comment · Reviewer_nP6r · 2024-08-09
> >
> > > (1) As discussed in ALL-Q1, while the vision encoder aids in early convergence, it becomes less crucial as the data scale increases. We introduce vision encoder distillation to improve training efficiency, especially with a limited data scale, which is less significant after 24M pre-training data in our experiments. This indicates that eliminating the vision encoder for a fully encoder-free VLM is practical during both training and inference.
> >
> > It's not convincing to me that "it becomes less crucial as the data scale increases". Empirically, vision encoder can be trained easily with noisy large scale web data (tens of billions); while generative VLM does require higher quality data (e.g. sentence like text to align with images), which makes it less easy to scale the VLM training data. I.e., "This indicates that eliminating the vision encoder for a fully encoder-free VLM is practical" might not be true.
> >
> > Could you please elaborate how "Scaling performance of EVE with or without vision encoder supervision" was conducted? E.g., what/how much data was used in which stage?
> >
> > > (2) The LLM-guided pre-aligning stage is one crucial factor for training stability and scaling efficiency.
> >
> > What is learned in this stage when "without vision encoder supervision"?
> >
> > > While maintaining language ability in stage 2 is a concern, it is not the primary challenge in developing encoder-free VLMs effectively.
> >
> > Have you tried maintaining language ability in stage 2 or 3?

---

> > > ### Author Response · Authors · 2024-08-11
> > > **New reply and clarification to reviewer nP6r**
> > >
> > > Thank you for considering our response and for your valuable feedback. We appreciate the opportunity to further address the concerns raised.
> > >
> > > `Q1: "It becomes less crucial as the data scale increases" is not convincing. VE can be trained easily with noisy large-scale web data; while generative VLM does require higher quality data, which makes it less easy to scale the VLM training data. I.e., "This indicates that eliminating the vision encoder for a fully encoder-free VLM is practical" might not be true.`
> > >
> > > **(1) In the context of the rebuttal, "*it* becomes less crucial ..." and "eliminating the *vision encoder* for a ..." refer specifically to the distillation/supervision of the vision encoder, not the vision encoder used in encoder-based VLMs.** In Figure 1 of the rebuttal PDF, vision encoder distillation does help early convergence with a moderate data scale, but its importance diminishes with larger data sets (beyond 24M in our experiments). This suggests that removing the vision encoder supervision in EVE is feasible with extensive pre-training data.
> > >
> > > **(2) Removing the vision encoder in encoder-based VLMs proves practical even with noisy large-scale data.** Only with 1B mixture data (including large noisy WebLI, CC3M-35L, and etc), Paligemma's encoder-free model nearly matches the encoder-based one (with 40B data for separate SigLIP) across about 13 of 37 datasets (e.g., ChartQA, GQA, RefCOCO/+/g, RSVQA, etc.). The scaling trend reflects the potential for further bridging their performance gap, indicating that a pre-trained vision encoder may not be essential and that using noisy data (especially interleave data [a]) to train generative VLMs is acceptable.
> > >
> > > **(3) We agree that high-quality data is crucial for VLMs compared to noisy data. However, scaling up high-quality data is not as difficult as it might appear.** For example, with LLaVA1.5-13B via the SGLang package, we can caption about 4-5M images in one day using two A100 (40G) nodes. With more devices, the process becomes even faster. Additionally, there are many open-source re-captioning data sources (e.g., CapFusion [b], Recap-Datacomp-1B [c]) and caption engines (e.g., ShareGPT4v [d], DenseFusion [e]) available for constructing pre-training data. Besides, in the text-to-image generation field, building large high-quality datasets is commonly developed in the research and industry. Thus, we do not view this as a significant bottleneck in real-world scenarios.
> > >
> > > [a] VILA: On Pre-training for Visual Language Models. CVPR2024
> > >
> > > [b] CapsFusion: Rethinking Image-Text Data at Scale. CVPR2024
> > >
> > > [c] What If We Recaption Billions of Web Images with LLaMA-3? arXiv 2406.08478
> > >
> > > [d] ShareGPT4V: Improving Large Multi-Modal Models with Better Captions. ECCV2024
> > >
> > > [e] DenseFusion-1M: Merging Vision Experts for Comprehensive Multimodal Perception. arXiv 2407.08303
> > >
> > > `Q2: Elaborate "Scaling performance of EVE with or without vision encoder supervision"`
> > >
> > > For fairness, we retain the Patch Embedding Layer (PEL) and remove the Patch Aligning Layer (PAL) (i.e., VE supervision) at all stages, keeping other variables unchanged.
> > >
> > > - Stage 1: We train both PEL and PAL for EVE-7B and EVE-7B-HD, and only PEL for versions without VE supervision, using 16M out of 33M pretraining data at the largest 672-pixel resolution.
> > >
> > > - Stage 2: We unfreeze all modules for these versions, utilizing all 33M pretraining data at the largest 672-pixel resolution.
> > >
> > > - Stage 3: We train the full model for EVE-7B (w/ and w/o VE supervision) using LLaVA-mix-665K, and for EVE-7B-HD (w/ and w/o VE supervision) using extra 1.2M SFT data at a 1344-pixel resolution.
> > >
> > > `Q3: What is learned in this stage when "without vision encoder supervision"?`
> > >
> > > Through text supervision, EVE aims to learn visual perception and establish an initial vision-language connection. This provides a better starting point for subsequent large-scale pretraining and avoiding model collapse.
> > >
> > > `Q4: Have you tried maintaining language ability in stage 2 or 3?`
> > >
> > > Yes, we conducted experiments using our multimodal data and text-only FineWeb data in 7:3, 5:5, and 3:7 ratios.
> > >
> > > - We used 5M mixed data for Stages 1 and 2, followed by LLaVA-mix-665K for Stage 3, all with a 1344-pixel image resolution, and removed vision encoder supervision throughout all stages.
> > >
> > > - We observed that a higher proportion of language-only data preserved better language capability (SQA scores improved from 63.5 to 64.7 to 65.2), but led to a slower increase in multimodal capability (GQA scores decreased from 58.3 to 57.4 to 55.2).
> > >
> > > - We suggest a 5:5 ratio for pre-training with high-resolution images. While we haven’t tested a 672-pixel resolution, a 7:3 (multimodal:text-only samples) ratio might be a good choice for balancing image-text token ratios, which is a better consideration factor.
> > >
> > > We respectfully hope that these explanations can convince you to potentially enhance your evaluation score. We are available to address any additional questions you may have.

---

> > > > ### Comment · Reviewer_nP6r · 2024-08-11
> > > >
> > > > Thanks for the quick reply! Partial my questions have been addressed. A few more below:
> > > >
> > > > > Only with 1B mixture data (including large noisy WebLI, CC3M-35L, and etc), Paligemma's encoder-free model nearly matches the encoder-based one (with 40B data for separate SigLIP) across about 13 of 37 datasets...
> > > >
> > > > PaliGemma paper says "Figure 8 (per-task breakdown in Appendix K.4) shows that while this architecture still significantly lags behind, the scaling with pretraining duration seems potentially promising.". It claims "significantly lags" (I also checked Appendix K.4), while you are saying "nearly matches". Is there any misleading here?
> > > >
> > > > > However, scaling up high-quality data is not as difficult as it might appear.
> > > >
> > > > I know this is out of the scope of this paper, just to mention I am not convinced by this statement. Empirically, synthetic captions indeed could help, but it can't beat the high quality image-text data due to less diversity and other issues.
> > > >
> > > > > For fairness, we retain the Patch Embedding Layer (PEL) and remove the Patch Aligning Layer (PAL) (i.e., VE supervision) at all stages, keeping other variables unchanged.
> > > >
> > > > Thanks for the details! I feel this paper could be more impactful if we could further explore the direction w/o PAL (which could be an ablation) due to the architecture's simplicity. Your current training stages look neat and make sense to me. It would be wonderful if finally the result w/o PAL could match the SOTA ones w/ vision encoder.
> > > >
> > > > > Yes, we conducted experiments using our multimodal data and text-only FineWeb data in 7:3, 5:5, and 3:7 ratios.
> > > >
> > > > This looks insightful to the community. Would you consider putting such ablations into the paper?
> > > >
> > > > One more question regarding patching: it is nice that EVE can accept arbitrary resolution and aspect ratio, which is normally not implemented in the pre-trained vision encoder. Have you tried image augmentations (such as simple resizing)? Which I feel helpful for further improving the performance.

---

> > > > > ### Author Response · Authors · 2024-08-13
> > > > > **New reply and clarification to reviewer nP6r**
> > > > >
> > > > > Thanks for your constructive feedback. Here are some explanations for the questions as follows:
> > > > >
> > > > > `Q1:<<Only with 1B mixture data (including large noisy WebLI, CC3M-35L, and etc), Paligemma's encoder-free model nearly matches the encoder-based one (with 40B data for separate SigLIP) across about 13 of 37 datasets.>> PaliGemma paper says "Figure 8 (per-task breakdown in Appendix K.4) shows that while this architecture still significantly lags behind, the scaling with pretraining duration seems potentially promising.". It claims "significantly lags" (I also checked Appendix K.4), while you are saying "nearly matches". Is there any misleading here?`
> > > > >
> > > > > We appreciate your careful examination of the PaliGemma paper. In our previous response, we mentioned that the encoder-free model "**nearly matches the encoder-based one across 13 of the 37 datasets**", specifically including datasets such as ChartQA, GQA, xGQA, ScienceQA, TallyQA, RefCOCO/+/g, RSVQA-hr, RSVQA-lr, OCR-VQA, ActivityNet-CAP, and MSRVTT-QA (shown in its Appendix K.4).
> > > > > Notably, PaliGemma's claim of "significant lag" refers to the overall performance averaged across all 37 datasets. Our early description may not contradict PaliGemma's statement. We hope this reply can address your question.
> > > > >
> > > > > `Q2: I know this is out of the scope of this paper, just to mention I am not convinced by this statement. Empirically, synthetic captions indeed could help, but it can't beat the high-quality image-text data due to less diversity and other issues.`
> > > > >
> > > > > Thanks for your insightful comments. We acknowledge that while web-scale or synthetic captions may introduce noise or lack diversity, they have proven valuable in scaling efficiency, as shown in PaliGemma and our EVE. However, we fully agree that high-quality image-text data remains the gold standard for training VLMs. Some strides have been made in academia, such as ShareGPT4V and DenseFusion, which leverage advanced vision experts and GPT-4v to create caption scaling systems. Efforts like LLaVA-OneVision and VILA2 further showcase the potential of refining training data in a closed loop, leading to large, high-quality datasets. In industry, major players like OpenAI, Meta, Google, etc are heavily investing in this aspect.
> > > > >
> > > > > These developments may help alleviate this problem, and we appreciate your recognition of the potential for future advancements.
> > > > >
> > > > > `Q3: Thanks for the details! I feel this paper could be more impactful if we could further explore the direction w/o PAL (which could be an ablation) due to the architecture's simplicity. Your current training stages look neat and make sense to me. It would be wonderful if finally the result w/o PAL could match the SOTA ones w/ vision encoder.`
> > > > >
> > > > > Thanks for your valuable feedback. We fully agree that exploring the model's performance without PAL is an interesting direction. As shown in Figure 1 of the Rebuttal PDF, its impact gradually decreases as data scale increases. By the 24M mark, the performance gap with and without supervision narrows to just 0.3-0.8\% across all datasets. We are confident that with training data further scaling up, EVE can be further simplified and the the performance without PAL may also rival the SOTA VLMs w/ vision encoder.
> > > > >
> > > > > `Q4: This looks insightful to the community. Would you consider putting such ablations into the paper?`
> > > > >
> > > > > We appreciate the suggestion and polish our manuscript accordingly.
> > > > >
> > > > > `Q5: One more question regarding patching: it is nice that EVE can accept arbitrary resolution and aspect ratio, which is normally not implemented in the pre-trained vision encoder. Have you tried image augmentations (such as simple resizing)? Which I feel helpful for further improving the performance.`
> > > > >
> > > > > Good question. We haven't explored image augmentations yet, which indeed can enhance performance. To push EVE's limits, we're now scaling up the training data, aligning with the philosophy of image augmentation, and potentially bringing further improvements in the future.
> > > > >
> > > > > We respectfully hope our reply addresses your concerns and potentially improves your evaluation score. We are available to provide further details or answer any additional questions you may have.

---

### Official Review · Reviewer_VqDP · 2024-07-11

**Soundness:** 2
**Presentation:** 3
**Contribution:** 2
**Rating:** 5
**Confidence:** 4

**Summary:**

Authors bridge the gap between encoder-based and encoder-free models, and present a simple yet effective training recipe towards pure VLMs. They unveil the key aspects of training encoder-free VLMs efficiently via thorough experiments: (1) Bridging vision-language representation inside one unified decoder; (2) Enhancing visual recognition capability via extra supervision

**Strengths:**

1. Authors propose to find the key aspects of training encoder-free VLMs
2. EVE outperforms the counterpart Fuyu-8B (encoder-free)

**Weaknesses:**

I am concerned with the motivation and experiments.

From Tab. 3, although it outperforms Fuyu-8B, it is inferior to many encoder-based Large Vision-Language Models. I would except EVE had additional benefits, however, it is unclear why we need it.

The AR is 'Any' for encoder-free VLMs, but I do not see the benefits of it. At least I cannot have it from the tables. Could authors provide any explanations? From Tab. 6, EVE-7B (HD), which had acceptable performance, does not gain enough benefits in terms of FLOPs(G) and Time(s) if we consider the vision part and LLM part together. At current stage, I could not see enough motivation of EVE-7B (HD).

One minor question (not urgent, authors could simply ignore it if time does not allow): the benchmarks presented are normally simple VQA benchmarks, how would the model perform on detailed description tasks?

Overall, I am concerned with the motivation.

**Questions:**

see weakness

---

> ### Author Rebuttal · Authors · 2024-08-07
>
> Thank you for your thoughtful remarks. We have responded to all your queries and made the necessary changes to enhance the paper.
>
> `Q1:  (((1))) EVE outperforms Fuyu-8B, but lags behind many encoder-based VLMs. why we need it?  (((2))) Any explanations for the benefits of 'Any image ratio'?  (((3))) No enough benefits of FLOPs(G) and Time(s) when considering the vision part and LLM part together.`
>
> **(1) We highlight that EVE shows potentially more promising scaling with pretraining duration, which is the key motivation behind our efforts to build encoder-free VLMs.**
> EVE attempts to remove strong vision inductive bias and transmit visual signals almost losslessly for better scaling properties.
> In Figure 2 of the rebuttal PDF, we observe that encoder-based models often suffer from collapse. Only the (VE)-(ALL) training strategy avoids this issue by freezing LLM weights during pre-training and unfreezing them during the SFT stage. In contrast, EVE shows better scaling properties and gradually approaches the performance of well-developed encoder-based VLMs with only 33M data scale.
>
> **(2) Encoder-free VLMs are promising for scaling but require enormous training data to develop vision perception from scratch.**
> Here, with only 33M pre-training data, our pioneering exploration currently lags behind but performs comparably to popular encoder-based methods. **Interestingly, subsequent PaliGemma [a]** also explores an encoder-free version via 1B pre-training image-text pairs, showing promising early results alongside its encoder-based counterpart across 37 validation datasets (see Figure 3 of the rebuttal PDF). They particularly mention that only the separate vision encoder i.e. SigLIP, has been trained with 40B image-text pairs, far greater than 1B data of encoder-free version. They also indicate that decoder-only VLMs may be a promising direction although currently suffering in training efficiency due to building vision perception from scratch.
>
> **(3) The 'Any' image ratio, simple architecture, and efficient deployment are bonuses of encoder-free VLMs.**
> Recent studies on encoder-based VLMs reveal that
> **(i)** Due to the limitations of pre-trained encoders, existing VLMs exhibit vulnerabilities in basic capabilities rooted in visual encoding trade-off [b, c].
> **(ii)** Various vision encoders show uneven levels of capability due to pretext pretraining tasks, relying heavily on the corrective capabilities of LLMs for multimodal understanding [d, e].
> In contrast, encoder-free VLMs remove semantic priors in abstracting visual representation, theoretically allowing VLMs to autonomously acquire all available information. **While 'any image ratio' and 'FLOPS gains' are natural benefits of the encoder-free approach, the primary reason for exploring an encoder-free model is its scaling efficiency with less inductive bias.**
> In this premise, removing the vision encoder provides only a modest bonus in terms of flexible image input and deployment efficiency. Notably, the encoder-free track is still in early development and has a long way to explore its limits.
>
> [a] PaliGemma: A versatile 3B VLM for transfer. Google DeepMind. arXiv 2407.07726.
>
> [b] LLaVA-UHD: an LMM Perceiving Any Aspect Ratio and High-Resolution Images. Xu et al. arXiv 2403.11703.
>
> [c] HiRes-LLaVA: Restoring Fragmentation Input in High-Resolution Large Vision-Language Models. arXiv 2407.08706.
>
> [d] Eyes Wide Shut? Exploring the Visual Shortcomings of Multimodal LLMs. Tong et al. CVPR2024.
>
> [e] Cambrian-1: A Fully Open, Vision-Centric Exploration of Multimodal LLMs. Tong et al. arXiv 2406.16860.
>
> `Q2: One minor question (not urgent, authors could simply ignore it if time does not allow): the benchmarks presented are normally simple VQA benchmarks, how would the model perform on detailed description tasks?`
>
> From the LMMs-Eval leaderboard [a], we observe that LLaVA-1.6 **surprisingly** achieves lower CIDEr scores than LLaVA-1.5 on COCO-Cap and Flickr-cap under the same model capacity. These description tasks do not well reflect the capability of multimodal models due to out-of-distribution issues and evaluation metrics, which are seldom evaluated by most existing VLMs.
>
> [a] LMMs-Eval: Reality Check on the Evaluation of Large Multimodal Models. Zhang et al. arXiv 2407.12772

---

> > ### Comment · Reviewer_VqDP · 2024-08-09
> >
> > Thanks for the detailed response. They addressed some of my concerns, although I still think it is over-claimed.
> >
> > For Q2, it is not surprising that LLaVA-1.6 achieves lower CIDEr scores than LLaVA-1.5 on COCO-Cap and Flickr-cap under the same model capacity, because LLaVA families are not designed (or trained) for concise image captioning. The improvements (data/model) made in LLaVA-1.6 are also not targeted at short image captioning. Therefore, I would not expect it can achieve better results than LlaVA 1.5.
> >
> > In fact, I am asking about the detailed description task (LLaVA Bench), where the model is asked to provide detailed description given an image (much longer and more comprehensive and diverse than image captioning). Have you tried the model with this dataset? It should be convenient as LLaVA already provided similar evaluation scripts.
> >
> > I would currently raise my score to 5

---

> > > ### Author Response · Authors · 2024-08-11
> > > **New reply and clarification to reviewer VqDP**
> > >
> > > Thank you for considering our response. We appreciate the opportunity to further clarify our contributions and address the concerns raised.
> > >
> > > `Q1: They addressed some of my concerns, although I still think it is over-claimed.`
> > >
> > > **(1) We respectfully reiterate that the key innovation of EVE is its preliminary validation of the feasibility of eliminating the inductive biases associated with encoder-based vision-language models (VLMs).** By removing these biases, EVE provides a clear path for constructing encoder-free VLMs that do not require an encoder pretrained on an intermediate task with biases in representations and resolutions. Thus, EVE maximizes the model's autonomy in learning vision perception and aligning multimodal patterns for better scaling efficiency, inspired by **'The Bitter Lesson' (Rich Sutton)**.
> > >
> > > **(2) Our EVE aligns with recent trends in VLM research, where the focus is shifting towards models that reduce architectural biases, construct a unified backbone, and improve scaling efficiency.** Notable examples of this trend are listed in time order:
> > >
> > > - Fuyu-8B: A Multimodal Architecture for AI Agents. Adept AI. October 17, 2023.
> > >
> > > - Chameleon: Mixed-Modal Early-Fusion Foundation Models. Meta FAIR. arXiv 2405.09818.
> > >
> > > - SOLO: A Single Transformer for Scalable Vision-Language Modeling. UIUC. arXiv 2407.06438.
> > >
> > > - PaliGemma: A versatile 3B VLM for transfer. Google DeepMind. arXiv 2407.07726. (Its encoder-free version)
> > >
> > > - MoMa: Efficient Early-Fusion Pre-training with Mixture of Modality-Aware Experts. Meta FAIR. arXiv 2407.21770.
> > >
> > > - GPT-4o System Card. OpenAI. August 8, 2024.
> > >
> > > These influential works exemplify a broader shift towards reducing architectural constraints in VLMs, aiming to develop end-to-end models that process textual and visual inputs through a unified network, thereby enhancing scaling efficiency.
> > >
> > > **(3) Our preliminary results demonstrate that EVE is not only feasible but also holds good promise for advancing VLM development.** We respectfully hope that this clarification would convince you to reconsider our proposed terminology and potentially improve your evaluation score based on the clarified context. We are open to providing further details or addressing any additional questions you may have.
> > >
> > > `Q2: In fact, I am asking about the detailed description task (LLaVA Bench), where the model is asked to provide a detailed description given an image (much longer and more comprehensive and diverse than image captioning). Have you tried the model with this dataset? It should be convenient as LLaVA already provided similar evaluation scripts.`
> > >
> > > Since LLaVA-Bench requires the GPT-4 API, we didn't include it in our earlier validation. We conducted further experiments using EVE-7B (HD) trained with LLaVA-mix-665 on LLaVA-Wide, achieving a 56.9% score, which is lower than LLaVA's 65.4%. This performance gap may be due to the QA-oriented SFT data and the limited vision perception learned from the current million-scale pretraining data.
> > >
> > > We appreciate you bringing attention to the detailed description task like LLaVA-Bench.
> > > Moving forward, we will focus on enhancing the model's capabilities by incorporating more training data and developing well-constructed instruction data.

---

### Official Review · Reviewer_gQSV · 2024-07-11

**Soundness:** 3
**Presentation:** 3
**Contribution:** 2
**Rating:** 6
**Confidence:** 4

**Summary:**

This paper explores the topic of encoder-free vision language model. It proposes to directly input the image patches into the decoder network together with the language tokens, without the use of a separate visual encoder during inference time. The benefits are mainly two-fold: simpler architecture and more flexible image resolution and aspect ratio. In experiments, they utilized 33M/1.8M data for pretraining/SFT and demonstrated superior performance to Fuyu-8B which is not open sourced.

**Strengths:**

1. The topic of encoder-free VLMs is very interesting and has valuable potential benefits.
2. As an empirical paper, the experimental results look solid. The performance almost matches the best of encoder-based VLMs . It is a plus that the proposed model outperforms the counterpart Fuyu-8B.
3. Experiments to verify the necessity of stage-1 training is properly done and provides some insight of training VLMs with added components.

**Weaknesses:**

1. The training is using an existing visual encoder as teacher, which kind of defeats the title of "encoder-free". Is this necessary? The contribution would be more valuable if it can be completely encoder-free (during both training and inference).
2. Related to 1., if there is a need to train the model on some ouf-of-domain datasets, does it need to first train the separate visual encoder, and then redo the 3 training stages described in this work? This makes the training even more complex than encoder-based VLMs.

**Questions:**

1. In Fig.6, what does “from scratch” mean exactly? Is the language model also randomly initialized?

**Limitations:**

Yes

---

> ### Author Rebuttal · Authors · 2024-08-07
>
> Thank you for your valuable comments. We respond to all questions you raise to address your concerns and make the necessary revisions to improve the quality of the paper.
>
> `Q1: The training is using an existing visual encoder as teacher, which kind of defeats the title of "encoder-free". Is this necessary? The contribution would be more valuable if it can be completely encoder-free (during both training and inference)`
>
> As discussed in ALL-Q1, we found that while the vision encoder aids in early convergence, it becomes less crucial as the data scale increases. Inspired by this, we empirically found that pre-training EVE with 4M high-resolution images (1344 longest edge) without vision supervision can match the performance of our original version, which used 12M data points and vision supervision. This indicates that eliminating the vision encoder for a fully encoder-free VLM is practical during both training and inference.
>
> `Q2: Related to 1., if there is a need to train the model on some out-of-domain datasets, does it need to first train the separate visual encoder, and then redo the 3 training stages described in this work? This makes the training even more complex than encoder-based VLMs.`
>
> Good question.
> Transferring the visual encoder initially may be beneficial due to its strong inductive bias and susceptibility to out-of-domain issues. Fortunately in our experiments, we discovered that after 24M pre-training data, vision encoder supervision becomes less essential for EVE to develop visual perception from scratch. This enables EVE to circumvent these issues by abandoning the vision encoder supervision and preserving the original three stages during the training process.
>
> `Q3: In Fig.6, what does “from scratch” mean exactly? Is the language model also randomly initialized?`
>
> Yes. "From scratch" denotes that the entire model is initialized randomly. We empirically discovered that training EVE from scratch faces extreme optimization challenges and usually suffers from model collapse.

---

### Author Rebuttal · Authors · 2024-08-07

To all reviewers:

We thank all reviewers for your constructive comments. We are encouraged by approvals on an **interesting and novel idea** (gQSV, nP6r, Z41h), **simple yet effective** (VqDP, nP6r, Z41h), **comprehensive experiment and solid results** (gQSV, nP6r, Z41h), and **insights for the VLM community** (gQSV, nP6r, Z41h). We conducted more experiments and analyses to address some reviewers’ concerns. The supplements in the attached PDF are summarized as follows:

- Scaling performance of EVE with or without vision encoder supervision in Figure 1 of the attached PDF.

- Scaling performance of EVE vs. the encoder-based baseline in Figure 2 of the attached PDF;

- Scaling performance of PaliGemma with or without an image encoder from 100M to 1B data in Figure 3 of the attached PDF;

`All Reply Q1: Is 'encoder-free' accurate? How does vision encoder distillation/supervision work in EVE? Is it necessary?`

**(1) Vision encoder supervision does help with early convergence by 1-3% gains shown in Table 4, but is not very crucial during large data scale-up.**
**(i)**  We introduce vision encoder supervision to improve training efficiency, especially with a limited data scale.
**(ii)**  Actually, vision supervision from a pre-trained encoder is less significant when using sufficient data resources.
Figure 1 of the rebuttal PDF indicates that the influence of vision encoder supervision diminishes over large data scales, and by the 24M mark, the difference in performance with or without this supervision is negligible by less than 0.3-0.8\%.
This may be because large amounts of high-quality and detailed captions greatly enhance the understanding of visual information, thus gradually reducing the need for visual encoders.

**(2) More importantly, vision supervision is not the crucial factor for training stability and scaling efficiency.**
Table 5 and Figure 6 show that even with vision supervision, performance without LLM-guided Pre-aligning in Stage 1 rapidly decreases as data volume increases beyond a certain point. This indicates that vision supervision is not essential for the scaling efficiency of EVE.

**(3) We came up with the ‘encoder-free’ concept because EVE can work like Fuyu-series without visual encoders during the inference and deployment.**
We can completely remove it during inference, allowing EVE to function as a pure encoder-free architecture like Fuyu-8B.
Besides, though involving vision supervision helps with training efficiency, it becomes less necessary as the pre-training data scale significantly increases. In other words, developing a fully encoder-free VLM during training or inference is practical with larger data and computing resources.

We hope that the point-by-point responses below have effectively addressed your previous concerns, and would appreciate any further feedback you can provide.

Sincerely yours,

Authors.

---

### Decision · Program_Chairs · 2024-09-25

**Decision:**

Accept (spotlight)

**Comment:**

The paper proposes to train LVLM without using pretrained vision encoders as an essential building block. It shows promising performance and the key aspects are simple and effective. This can be insightful to the research community to build better encoder-free MLLMs. The encoder free model can  offer flexible resolution and aspect ratio, improved efficiency. Concerns raised by reviewers were addressed during rebuttal. We encourage the authors to include the clarifications and discussion of main rebuttal points, and clarify the purpose of encoder free and lossless terms.